# Effects of Long-Term (17 Years) Nitrogen Input on Soil Bacterial Community in Sanjiang Plain: The Largest Marsh Wetland in China

**DOI:** 10.3390/microorganisms11061552

**Published:** 2023-06-10

**Authors:** Zhenbo Chen, Chi Zhang, Zhihong Liu, Changchun Song, Shuai Xin

**Affiliations:** Faculty of Infrastructure Engineering, Dalian University of Technology, Dalian 116023, China; chenzhenbo333@163.com (Z.C.); zhliu@dlut.edu.cn (Z.L.); songcc@dlut.edu.cn (C.S.); shxinizg@163.com (S.X.)

**Keywords:** soil microorganisms, long term, nitrogen input, nitrogen cycle, wetland

## Abstract

Increased nitrogen (N) input from natural factors and human activities may negatively impact the health of marsh wetlands. However, the understanding of how exogenous N affects the ecosystem remains limited. We selected the soil bacterial community as the index of ecosystem health and performed a long-term N input experiment, including four N levels of 0, 6, 12, and 24 gN·m^−2^·a^−1^ (denoted as CK, C1, C2, and C3, respectively). The results showed that a high-level N (24 gN·m^−2^·a^−1^) input could significantly reduce the Chao index and ACE index for the bacterial community and inhibit some dominant microorganisms. The RDA results indicated that TN and NH_4_^+^ were the critical factors influencing the soil microbial community under the long-term N input. Moreover, the long-term N input was found to significantly reduce the abundance of *Azospirillum* and *Desulfovibrio*, which were typical N-fixing microorganisms. Conversely, the long-term N input was found to significantly increase the abundance of *Nitrosospira* and *Clostridium_sensu_stricto_1*, which were typical nitrifying and denitrifying microorganisms. Increased soil N content has been suggested to inhibit the N fixation function of the wetland and exert a positive effect on the processes of nitrification and denitrification in the wetland ecosystem. Our research can be used to improve strategies to protect wetland health.

## 1. Introduction

Wetland is a fragile ecosystem. Despite the importance of wetland in supporting a wide range of ecosystem services such as water quality improvement, biodiversity, and climate regulation [1], it has been damaged by natural factors and human activities [2,3,4]. Sanjiang Plain is the largest swamp area in China and is also representative of the global temperate wetland ecosystem. Since the 1950s, as a result of several large-scale agricultural developments, the nutrient input and biodiversity of the wetland ecosystem have greatly changed due to high-intensity drainage and reclamation activities [5,6]. The annual TN loss load of paddy field in Sanjiang Plain is estimated to reach 25.3 kg/hm^2^, accounting for 15.3% of the total fertilization. Moreover, with the increasing use of fertilization, N leaching is bound to increase in severity, which may result in various potential damages to the ecosystem structure, function, and diversity of wetland [7,8]. Thus, the impact of long-term N input on the health of the entire wetland ecosystem requires an assessment.

To assess the change in wetland health, the evaluation target should be scientifically selected. Microorganisms in wetland soils are the basic producers and decomposers that sustain wetland functions and occupy very important ecological niches in wetlands. Microorganisms play important roles in regulating the biogeochemical cycle of wetland soil, such as carbon (C) and N-transforming networks [9,10]. Different microbiota can play different functions, such as methane oxidation, organic matter decomposition, and denitrification, which may determine the fate of C and N in wetlands [11,12,13]. Therefore, studying the structure and function of the microbial community has great benefits for assessing wetland health.

Additionally, as the most common limiting nutrient, the exogenous N input has been shown to significantly affect the function and composition of soil microorganisms in ecosystems. Research has shown that the characteristics of soil microorganisms such as bacterial and fungal composition [14] and total microbial biomass [15] can be affected by long-term N fertilization, which changes the N transformation process of soil. In peatland ecosystems, long-term N inputs increased the abundance and diversity of bacteria and fungi in the soil [16], but different results were obtained in forest ecosystems, which may be related to ecosystem and seasonal differences [17]. Numerous studies have demonstrated that a moderate range of N input can promote microbial growth [18]. However, excessive inorganic N input leads to soil acidification, thereby changing the structure of the soil bacterial community and reducing the fungal diversity related to soil ecological function [19,20]. Some researchers have also indicated that microbial growth was inhibited by N addition due to soil pH changes and plant impacts [21,22].

However, our knowledge of changes in soil microorganisms in marsh wetland under long-term exogenous N input, especially changes in some key processes of the material cycles, such as C and N cycle, is still limited. Whether a threshold exists for the N input level beyond the critical transitions of microbial communities also remains unclear. To illustrate the effects of long-term N input on the structure and function of microorganisms in marsh wetland ecosystems, we conducted in situ N input gradient experiments with four different nutrient input treatments. The following research hypothesis was formulated: the community structure of soil microorganisms is strongly influenced by exogenous N inputs, and changes in soil microbial function are an adaptive strategy in response to environmental changes. The aims of this study were to (1) reveal the changes in soil bacterial diversity and community structure of swamp wetland ecosystem; (2) determine the impact on key processes and functional microorganisms involved in N cycle under long-term exogenous N input; and (3) identify the critical environmental factors affecting the structure and function of soil bacteria. Our study can provide a partial theoretical basis for the ecological restoration of marsh wetlands under long-term N pollution.

## 2. Materials and Methods

### 2.1. Experimental Design and Sample Collection

Sanjiang Plain is the largest swamp distribution area in China. Sanjiang Plain swamp wetland ecological test station is located in Honghe farm in Jiamusi City, Heilongjiang Province, in the northeast of Sanjiang Plain (47°35′ N and 133°31′ E). Due to atmospheric N deposition and drainage from agricultural fields, the N content in the Sanjiang wetlands has increased. Hygrophytic and biogenic plants primarily include *Elaeagnus microphylla*, *Salix rosmarinifolia*, *Carex* spp., and *reed*. The soil type in the study area belongs to meadow soil among the semi-hydromorphic soil based on the Chinese Soil Taxonomy (CST) [23]. The annual precipitation in the Sanjiang Plain is approximately 600 mm, with about 80% of it centered from June to October.

In the present study, four N treatment levels of 0, 6, 12, and 24 gN·m^−2^·a^−1^ (denoted as CK, C1, C2, and C3, respectively) were set, with three repetitions for each treatment (Figure 1), and the natural N input level was comparable to C1 [24]. A 1 m × 1 m iron sheet was surrounded to prevent lateral N loss, and a 1 m isolation area was placed between each community. For the experimental area used, to better simulate the process of agricultural nitrogen fertilization into wetlands, NH_4_NO_3_ has been continuously applied since 2005 in the growing season (May to September), and N has been sprayed on the experimental plot in the form of solution every month.

The soil samples were sampled in early September 2021. A detailed description of the experimental sites is available in the study of Mao and Ding [25,26]. Since soil nutrients and microbial biomass generally decrease with depth [27,28], and the subsoil of 0–10 cm is occupied by plant roots, we selected the 10–20 cm soil layer as the sampling area. In a typical procedure, three soil samples (diameter of 6 cm and depth of 10–20 cm) were randomly collected from each test plot with a hand drill and then evenly mixed to form a composite sample. A portion of the sample was air-dried and milled to pass through a 2 mm mesh sieve for further analysis of soil properties. The remainder was sent to the lab with ice packs in a cooler and kept at −80 °C for soil microbiological analysis.

### 2.2. Soil Physicochemical Properties and Vegetation Investigation

The contents of N (NH_4_^+^, NO_3_^−^, NO_2_^−^, and TN) in soil samples were measured with a continuous-flow analyzer (Seal Analytical, Norderstedt, Germany). pH was measured by the electrode method. About 10 g of air-dried soil sample was placed into a 50 mL beaker, followed by 25 mL of ultrapure water, which was then mixed with a stirrer for 1 min and measured after 30 min. The conductivity was measured with a conductivity meter. First, 10 g of air-dried soil was placed in a 50 mL triangular flask, followed by 100 mL of ultrapure water. After 5 min of oscillation, the sample was filtered and measured with a conductivity meter. The average heights of plants, belonging to *Carex* spp., of the 12 sample plots were also measured and recorded in the table.

### 2.3. DNA Extraction and PCR Amplification

The genome DNA of soil microorganisms was derived from 12 samples with the E.Z.N.A. Soil DNA Kit (Omega Bio-tek, Norcross, GA, USA). DNA extracts were examined on 1% agarose gel. The purity of DNA was measured using NanoDrop 2000 UV–Vis spectrophotometer (Thermo Scientific, Wilmington, DE, USA).

With an ABI GeneAmp^®^ 9700 PCR thermocycler (ABI, Carlsbad, CA, USA), the universal 16S rRNA gene primer pairs 338f (5′-ACTCCTACGGAGCA-3′) and 806r (5′-GGACTACHVGGTWTCTAAT-3′) were selected for PCR product amplification. The PCR mixture was 20 µL, including 4 µL FastPfu buffer (5 × TransStart), 0.8 μL forward primer (5 μM), 0.8 μL reverse primer (5 μM), 2 µL dNTPs (2.5 mM), 10 ng template DNA, 0.4 μL TransStart FastPfu DNA polymerase, and ddH2O. The amplification procedure was as follows: predenaturation at 95 °C for 3 min, 27 cycles (denaturation at 95 °C for 30 s, annealing at 55 °C for 30 s, extension at 72 °C for 30 s), followed by stable extension at 72 °C for 10 min, and final storage at 4 °C. The PCR products were withdrawn from agarose gels (2%), purified with an AxyPrep DNA Gel Extraction Kit (Axygen Biosciences, Union City, CA, USA), and quantified with a Quantus™ Fluorometer (Promega, Madison, WI, USA).

### 2.4. Illumina MiSeq Sequencing

High-throughput sequencing was performed with the Illumina MiSeq platform (Miseq PE300/NovaSeq PE250) at the Shanghai Majorbio Bio-pharm Technology Co., Ltd. (Shanghai, China). Raw reads were stored in the NCBI Sequence Read Archive database under the BioProject accession numbers PRJNA890032.

### 2.5. Quantitative Real-Time PCR

Following the DNA extraction process, bacterial abundance was measured using RT-PCR on an ABI 7300 instrument (Applied Biosystems, Carlsbad, CA, USA). The reaction components included 10 µL of 2X ChamQ SYBR Color qPCR Master Mix, 0.4 μL of 50X ROX Reference Dye 1, 0.8 μL of forward primer (5 μM), 0.8 μL of reverse primer (5 μM), 6 µL of ddH_2_O, and 2 µL of template DNA, making a total of 20 µL. The cyclic conditions for the reaction were as follows: 95 °C for 3 min, followed by 40 cycles of 95 °C for 5 s, 58 °C for 30 s, and 72 °C for 1 min. The R^2^ value of the standard curve was 0.9974, and the melt curve had a single peak. RT-PCR was repeated threefold per soil sample.

### 2.6. Processing of Sequencing Data

The raw sequences were filtered for quality using fastp (version 0.20.0) [29] and spliced with FLASH (version 1.2.7) [30]. The procedures involved quality filtering and splicing of bases, screening for non-compliant sequences, distinguishing samples, and adjusting sequences accordingly. For a more detailed explanation, please refer to Ma and Yang’s study [31,32].

In this subject paper, UPARSE (version 7.1) was used to group operational taxonomic units and exclude chimeras based on a 97% similarity threshold [33]. With the RDP classifier, each sequence was annotated taxonomically [34] based on the Silva 16S rRNA database (v138) [35] with a 70% comparison threshold.

### 2.7. Statistical Analyses

Alpha diversity indices (Sobs, Chao1, ACE, Shannon, Simpson) representing species richness and diversity were calculated with QIIME (version 1.9.1). Coverage index reflected the coverage of soil bacteria. The variability among samples was determined by principal coordinate analysis. Similarities and differences between different N input groups at the OTU level were determined by analysis of similarities and nonmetric multidimensional scaling (NMDS), with both distance algorithms using abund_jaccard. RDA served to determine the relation of each sample to environmental factors. Correlation heatmap was used to show the correlation between different groups of bacteria in the samples and environmental factors based on the Pearson correlation coefficient. Sequencing data were analyzed using the Majorbio I-Sanger Cloud Online Platform.

We combined high-throughput sequencing with quantitative PCR to obtain the absolute number of functional microorganisms [36,37]. Differences between treatments were evaluated using one-way ANOVA and least significant difference test (SPSS 26.0 IBM, Chicago, IL, USA) with a significance level of α = 0.05. Correlation plots between environmental factors and soil N cycling functional flora were produced using Originpro (version 2022b, Northampton, MA, USA) based on Pearson correlation coefficients to illustrate the influence of environmental factors on N cycling processes.

## 3. Results

### 3.1. Diversity and Composition of Microbial Community

Figure 2a shows the alpha diversity index of soil bacteria in this experiment. The ACE index and Chao index were basically unchanged at low-level N (C1), but decreased significantly at high-level N input (C3) in contrast to the control group, indicating that a high N input could significantly reduce the richness of the soil bacterial community. The changes in the Shannon index and Simpson index under the N input indicated that there were no significant differences in the soil bacterial diversity of different experimental variants. The coverage was as high as 0.98, suggesting that the test results were representative of the real condition of microorganisms in the sample. Thus, high-level, long-term N input had a significant stress effect on soil bacteria, thereby reducing microbial community richness and being selective for soil bacteria.

Under long-term different N input levels, the community structure of wetland soil bacteria varied at the phylum and genus levels. Figure 2b shows that the main phyla of bacteria were *Actinobacterita*, *Proteobacteria*, *Firmicutes*, *Acidobacterita*, *Chloroflex*, *Bacteroidota*, *Desulfobacterota*, and *Myxococcota*. These dominant bacteria did not change significantly under different levels of long-term N inputs, indicating that they were relatively stable and adaptive to environmental changes. The abundance of *Spirochaetota* and *Dependentiae* did not change significantly under low-level N inputs but decreased significantly under high-level N inputs. Long-term, high-level N input was selective for some wetland soil bacterial species.

Figure 2c shows that the main genera of bacteria were *Bacillus*, *norank_f_Bacteroidetes_vadinHA17*, *Candidatus_Udaeobacter*, *norank_f_Xanthobacteraceae*, *Acidothermus*, *Aquisphaera*, *Oryzihumus*, and *Conexibacter*. Among them, *Conexibacter* are strictly aerobic bacteria [36], indicating that wetland soils remained in an aerobic environment and were in contact with dissolved oxygen at 10–20 cm. The abundance of *norank_f_norank_o_Gaiellales*, *Roseiarcus*, *Candidatus_koribacter*, *Syntrophobacter*, and other bacteria decreased significantly under long-term N inputs, indicating that these bacteria were sensitive to environmental changes. *Clostridium_sensu_stricto_1*, *Rhodoblastus*, *norank_f_Spirochaetaceae*, *unclassified_f_Rhodocyclaceae*, and *norank-f_Lentimicrobiaceae* showed a significant increase in abundance under long-term N input compared with the control, suggesting that N stress favored an increase in these bacteria’s abundance. The dominant strains of *Orquilmus* and *Aquidium* were not sensitive to N stress. In summary, N stress had a significant effect on the wetland soil microbial community.

### 3.2. Similarities and Differences of Microbial Community Structure

The results of a similarity and difference analysis in the microbial community structure are shown in Figure 3a,b. Different colored graphics represent samples with different levels of N input. At the phylum level, the two principal component axes explained 60.21% of the sample differences, with a significance of 0.013 (<0.05), indicating a significant difference between the four samples. At the genus level, the two principal component axes explained 61.82% of the variance in sample composition, with a significance of 0.001 (<0.05), indicating a significant separation of samples at different levels of N input. Figure 2a shows a high intragroup similarity between the C1 and C2 treatment groups. The different levels of long-term N input had significant effects on the wetland soil bacterial community structure, more significantly at the bacterial genus level.

Figure 3c shows the smallest intragroup differences in CK and the largest in C3. The differences among treatment groups were significantly greater than those inside groups, indicating significant differences in the bacterial communities between treatment groups at the outer level. Figure 3d shows that the stress function value of the soil sample was less than 0.1 with R^2^ = 0.5981 (*p* = 0.002 < 0.05), indicating that the NMDS assessment was accurate. The bacterial communities of the low (C1)- and medium (C2)-level N treatment groups were similar but significantly different from those in the CK and high (C3)-level N treatments. Thus, long-term N input had a significant impact on the bacterial community, and the degree of impact varied with increased N input.

### 3.3. Effects of Different Environmental Factors on Microorganisms

The characteristics of soil pH, TN, NH_4_^+^, NO_3_^−^, NO_2_^−^, σ, and plant height are shown in Figure 4A. The results showed an increase in the content of various N in the soil by N application. Compared with CK, the soil NH_4_^+^ content increased significantly in C1, whereas the content of NO_2_^−^ increased significantly in C2 and increased continuously with increased N addition. Furthermore, the content of TN in C3 was significantly higher than that in CK. The NO_3_^−^ content increased after C1 and C2 and decreased after C3 treatments. Different from the soil N content, the pH decreased with increased N levels, but the change was not significant. Soil conductivity (σ), which reflects salinity, increased and then decreased, but the differences between treatment groups were not significant. Plant height decreased significantly with increased N input, indicating that N input may also have a stress effect on the plants.

We conducted RDA on soil bacterial community and environmental factors at the genus and phylum levels to reveal the correlations of environmental factors with microbial community in soil. Figure 4B shows that at the genus level, the total interpretation rate of environmental factors on the change in bacterial community was 46.85%, with TN, NH_4_^+^, and height being highly associated with bacterial community changes. Figure 4B shows no significant correlation between the bacterial community and NO_3_^−^, NO_2_^−^, pH, and σ (*p* > 0.05). At the phylum level, environmental factors could explain 52.34% of the bacterial community variation (Figure 4C). Different from the genus level, TN had the highest correlation among environmental factors (R^2^ = 0.609), and NO_2_^−^ had a significant correlation (R^2^ = 0.468, *p* ≤ 0.05) (Figure 4C).

### 3.4. Effects on N Cycling Microorganisms

Bacteria involved in the soil effects on N cycling microorganisms and plant pathogenic microorganisms cycle include N-fixing, nitrifying, and denitrifying bacteria. *Serratia*, *Bradyrhizobium*, *Curtobacterium*, *Azospirillum*, *Desulfovibrio*, and *Rhizobacter* are N-fixing bacteria [38,39,40]. Figure 5A demonstrates their relative abundance and absolute numbers with N input. *Serratia* and *Curtobacterium* appeared only in C1 and C3, indicating that the addition of N addition the survival of both genera. The relative abundance of *Azospirillum* and *Desulfovibrio* showed significant decreases at medium and high levels of N, but no significant changes in the absolute number of bacteria were observed at high N inputs. *Bradyrhizobium* had the highest abundance and number and did not change significantly under N input, indicating its robustness to N stress. *Rhizobacter* decreased in abundance and number under N input, but did not change significantly.

The Nitrobacter detected included *Nitrosospira* and *Nitrosomonas*, the main genera of ammonia-oxidizing bacteria in soil [41]. *Nitrosospira* and *Nitrosomonas* appeared only with N input, and the abundance and number of *Nitrosospira* increased with N addition, suggesting that the N addition had a promotive effect on their growth. *Clostridium_sensu_stricto_1*, *Clostridium_sensu_stricto_12*, and *Clostridium_sensu_stricto_13* are denitrifying bacteria that can convert NO_3_^−^ to N_2_ [42,43]. *Conexibacter* can reduce NO_3_^−^ to NO_2_^−^ [44]. The abundance of *Clostridium_sensu_stricto_1* showed a significant increase with N addition, but the number did not change significantly. The abundance of *Clostridium_sensu_stricto_12*, *Clostridium_sensu_stricto_13*, and *Conexibacter* was higher and did not change significantly with N addition, indicating that the exogenous N input had less effect on the denitrifying bacteria.

We investigated the relationship between environmental factors and soil N cycling bacteria by correlation analysis. Figure 5B,C shows that *Desulfovibrio* was highly correlated with environmental factors and negatively correlated with N content, indicating that the increase in soil N had a stressful effect on some N-fixing bacteria. *Nitrosospira* showed less correlation with the soil N content but a significant negative correlation with the plant height. The relative abundance of *Clostridium_sensu_stricto_13* showed significant positive correlations with TN, NO_3_^−^, and NO_2_^−^ (Figure 5B), indicating that increased NO_3_^−^ and NO_2_^−^ content promoted denitrification processes in the soil. *Conexibacter* showed significant negative correlations with TN, NH_4_^+^, and NO_2_^−^ (Figure 5B), indicating that the coercive effect of N input was greater than the facilitative effect.

## 4. Discussion

### 4.1. Effects of N Input on Major Functional Microorganisms and Bacterial Diversity

The ecological functions of wetland ecosystems are closely associated with the microbial community composition of their unique habitats [45,46]. Soil microorganisms have various ecological functions and are extensively involved in wetland material cycling processes, such as C and N conversion [47,48,49]. Some dominant genera in Sanjiang wetlands are involved in soil N fixation and denitrification processes, such as *Bradyrhizobium*, *Paenibacillus*, and *Clostridium_sensu_stricto_13* [42,50,51], which play important roles in N transformation in wetlands. *Conexibacter* and *norank_f_Geobacteraceae* were involved in soil C cycling [52,53] and significantly decreased under high N inputs, suggesting that an exogenous N input was inhibitory to some C-cycling processes in the ecosystem [54]. This finding differs from some of the existing research [55,56]. *Candidatus_Solibacter* showed a significant decrease in abundance with N input, which is the main genus of bacteria for the remediation of heavy-metal contamination in soil, indicating that N addition inhibited its growth. These changes in functionally important dominant microorganisms may be an applicable strategy to environmental alterations in wetlands.

Soil microbial diversity reflects variations of the microbial community composition. Many studies have been conducted on the impact of the N input on the diversity of soil microorganisms [57,58]. For soil microorganisms that use N compounds as a nutrient or the energy source, the N input can have a direct effect on them [59]. For other microorganisms, the N input can further affect their growth indirectly through plant physiology and soil physicochemical properties [57]. Wei [60] noted that anthropogenic disturbances commonly reduce the microbial diversity of freshwater wetlands. However, the effect of N addition on bacterial diversity may be positive or negative depending on the concentration and type of N added [61]. This study showed that the richness of wetland soil bacteria responded nonlinearly to long-term N addition; specifically, it was unaffected by N input below 12 gN·m^−2^·a^−1^, but decreased prominently when the N input reached 24 gN·m^−2^·a^−1^. This phenomenon suggested that although the wetland bacterial communities were resistant to medium- and low-level N inputs, further increases in the N input may exceed the threshold and cause the bacterial community to shift to a low-diversity state.

### 4.2. Changes in Soil Environmental Factors under Long-Term N Inputs and Their Effects on Microorganisms

Soil is an important site for material exchange and physicochemical reactions in wetland ecosystems. Different concentrations of nutrients entering wetlands can cause an impact on soil properties and microbial communities [62]. Long-term exogenous N input significantly enhanced the content of various types of N (TN, NH_4_^+^, NO_3_^−^, and NO_2_^−^) and improved nutrient availability in wetland soils, consistent with the results of Qu in a medium-term N input experiment [63]. We found that the N contents of different treatment groups were not strictly multiplicative, which may be due to exogenous N inputs that alter the N cycling processes, including N fixation and N emission in the ecosystem [64,65]. Studies also showed that the total N of wetlands decreased under long-term N input, which may be related to environmental conditions and N addition methods [66]. The soil of Sanjiang wetland showed weak acidity (5.21–5.31), and pH gradually decreased with N addition. Mou [67] and Sui [68] obtained similar results in a short-term N-application experiment.

Soil physicochemical factors and vegetation are important environmental factors affecting wetland microbes [69,70]. Many researches have demonstrated that pH is an essential factor influencing soil microorganisms [71,72], and that the optimum pH values exist for specific microbial strains [73]. However, the range of pH variation in this study was small, so it was not a major factor affecting the variability of the bacterial community. Cai et al. [74] noted that NH_4_^+^ and TN can have important effects on the microbial community structure. In the current study, TN and NH_4_^+^ in soil were significantly correlated with changes in the microbial structure, suggesting that nutrients such as TN and NH_4_^+^ were the key factors influencing wetland soil microorganisms.

### 4.3. Effects of N Input on Soil N Cycling Microorganisms in Wetland

Environmental changes due to the N input can affect key microorganisms in the N cycle in different directions and to different degrees [16,75], which are related to the microbial species, the timing and amount of N application, and the ecosystem type [76,77]. Many microorganisms involved in N fixation are heterotrophic or mixed nutrients [78], so adding organic matter is often beneficial to N-fixing bacteria [79]. Nevertheless, the response of N-fixing bacteria to inorganic N addition is controversial [80,81]. N application reportedly does not affect or have a negative effect on soil microbial N fixation [82]. In the present study, the N input reduced the abundance and numbers of *Azospirillum* and *Desulfovibrio* (Figure 6), indicating that the stressing effect of N addition on diazotrophic taxa is higher than the positive effect brought by nutrient improvement [83]. The abundance of *Rhizobacter* gradually declined with N addition and showed a negative correlation with various N species (Figure 6). This finding indicated that the N input reduced the N fixation function in soil [84]. pH has been shown to significantly affect diazotrophic taxa [85], but in this study, the results differed, and the soil pH was found to be weakly correlated with the abundance of diazotrophic taxa. Overall, exogenous N input inhibited the N_2_ fixation by wetland microorganisms, which was probably due to the increased N in the soil [86], thereby reducing the competitive advantage of N-fixing bacteria [87].

In this study, *Nitrosospira* and *Nitrosomonas* significantly increased under long-term N input and were positively correlated with N, indicating that increased soil N favored the growth of nitrifying bacteria (Figure 6). However, the diversity of anaerobic ammonia-oxidizing bacteria in sediments was negatively correlated with NH_4_^+^, which is probably associated with the adaptation of different bacteria to the NH_4_^+^ concentration and the surrounding environment [75]. Studies have shown that pH is the primary factor leading to differences among bacterial community composition [88,89,90] and that it significantly affects the nitrification rate in wetland sediments [91]. Bouskill [92] also noted that ammonia availability is one of the determinants of nitrifying bacterial diversity and can be strongly influenced by pH. The present study had different results in that the abundance of *Nitrosospira* and *Nitrosomonas* under N input was not significantly correlated with NH_4_^+^ and pH, which may be due to the insignificant variation in the soil pH. Studies have found variability in the response of soil canonical nitrifiers to N addition at different soil depths, which may be a matter of interest in subsequent wetland studies.

Denitrification processes in the soil N cycle are primarily mediated by denitrifying microorganisms, which play essential roles in controlling the destination of N [93]. Prior studies have shown that denitrifying microorganisms are influenced by various environmental factors and are sensitive to changes in soil physicochemical properties [94,95,96]. In the present study, we found that long-term exogenous N addition enhanced the abundance and number of some denitrifying microorganisms, which were positively correlated with various N contents in soil. Several other studies have yielded similar results [97]. However, Chen [95] studied three different natural wetland habitats and concluded that denitrifying bacterial abundance decreases with increased ammonium and total N concentrations, which may be related to the different species of denitrifying microorganisms. The abundance of *Conexibacter* in this study decreased with the N addition (Figure 6), which also indicated that the stability of different species of denitrifying microorganisms differed in response to environmental changes [98,99,100]. In general, long-term exogenous N input exerted a facilitative effect on denitrification processes and accelerated N removal from wetlands [101,102,103].

## 5. Conclusions

Our study revealed changes in the structure and function of wetland microbial communities under long-term exogenous N input. Long-term, high-level N inputs could significantly reduce the ACE and Chao indices of soil bacteria and were selective for soil bacteria. Wetland microbial communities were resistant to medium- and low-level N inputs, but high-level N input was significantly inhibitory to microbial communities. Exogenous N input could significantly increase soil N (TN, NH_4_^+^, NO_3_^−^, and NO_2_^−^) contents, with TN and NH_4_^+^ being the key environmental factors affecting soil microbial community structure. We found that long-term exogenous N input can have an impact on the ecological functions of wetland soil, such as inhibiting N fixation processes and promoting nitrification and denitrification processes in soil.

## Figures and Tables

**Figure 1 microorganisms-11-01552-f001:**
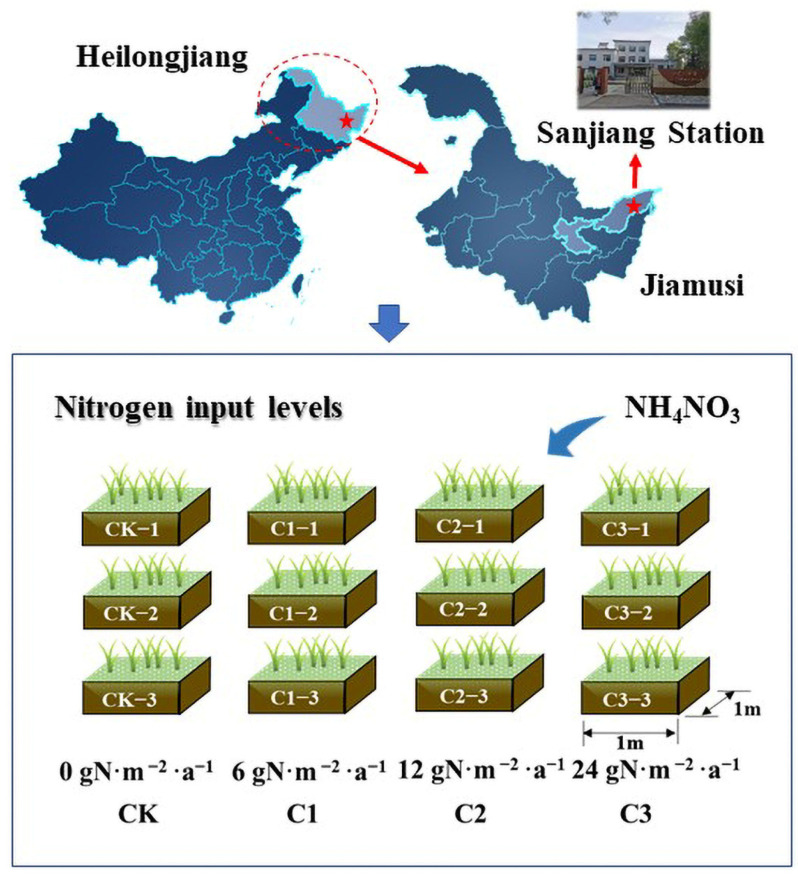
Location of the study area and experimental design. CK: 0 gN·m^−2^·a^−1^, including CK−1, CK−2, and CK−3; C1: 6 gN·m^−2^·a^−1^, including C1−1, C1−2, and C1−3; C2: 12 gN·m^−2^·a^−1^, including C2−1, C2−2, and C2−3; C3: 24 gN·m^−2^·a^−1^, including C3−1, C3−2, and C3−3. “g m^−2^ a^−1^” means the mass of nitrogen added per year per square meter.

**Figure 2 microorganisms-11-01552-f002:**
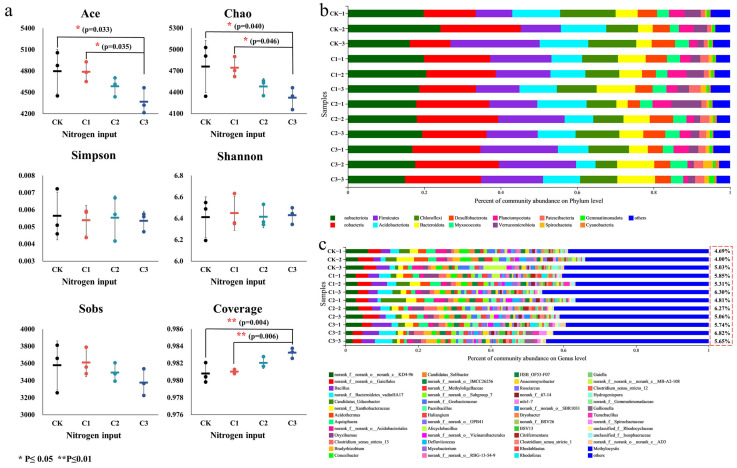
(**a**) Alpha diversity of soil bacteria. CK: 0 gN·m^−2^·a^−1^; C1: 6 gN·m^−2^·a^−1^; C2: 12 gN·m^−2^·a^−1^; C3: 24 gN·m^−2^·a^−1^. Error bars are based on the standard deviation of the means (*n* = 3). (**b**) Community composition of soil bacteria at phylum level. Others: relative abundance of bacterial phylum less than 1%. (**c**) Community composition of soil bacteria at genus level. Others: relative abundance of bacterial genus less than 1%. The numbers are the percentage of unknown microorganisms at genus level.

**Figure 3 microorganisms-11-01552-f003:**
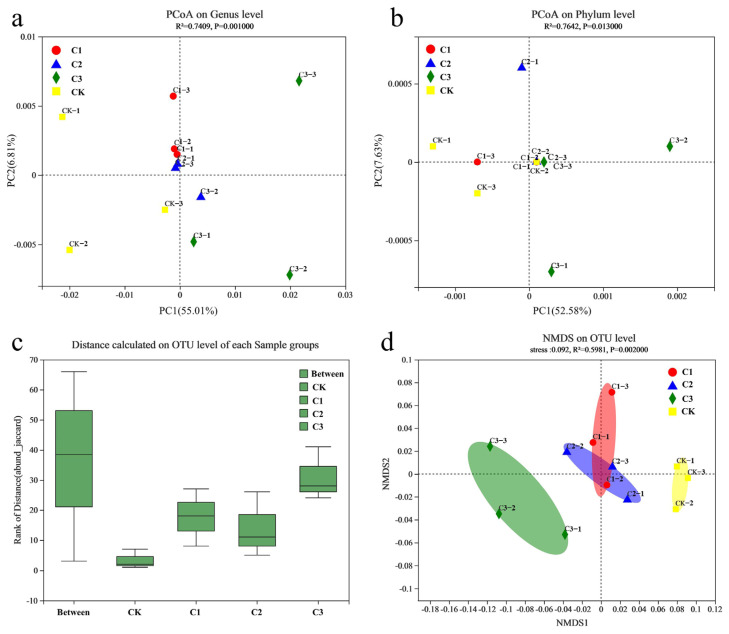
(**a**) Principal Co-ordinates Analysis (PCoA) of bacterial communities under genus level. (**b**) Principal Co-ordinates Analysis (PCoA) of bacterial communities under phylum level. (**c**) Analysis of similarities (ANOSIM) of bacterial communities under OTU level. (**d**) Non-metric multidimensional scaling (NMDS) of bacterial communities under OTU level. CK: CK−1, CK−2, CK−3, and 0 gN·m^−2^·a^−1^; C1: C1−1, C1−2, C1−3, and 6 gN·m^−2^·a^−1^; C2: C2−1, C2−2, C2−3, and 12 gN·m^−2^·a^−1^; C3: C3−1, C3−2, C3−3, and 24 gN·m^−2^·a^−1^.

**Figure 4 microorganisms-11-01552-f004:**
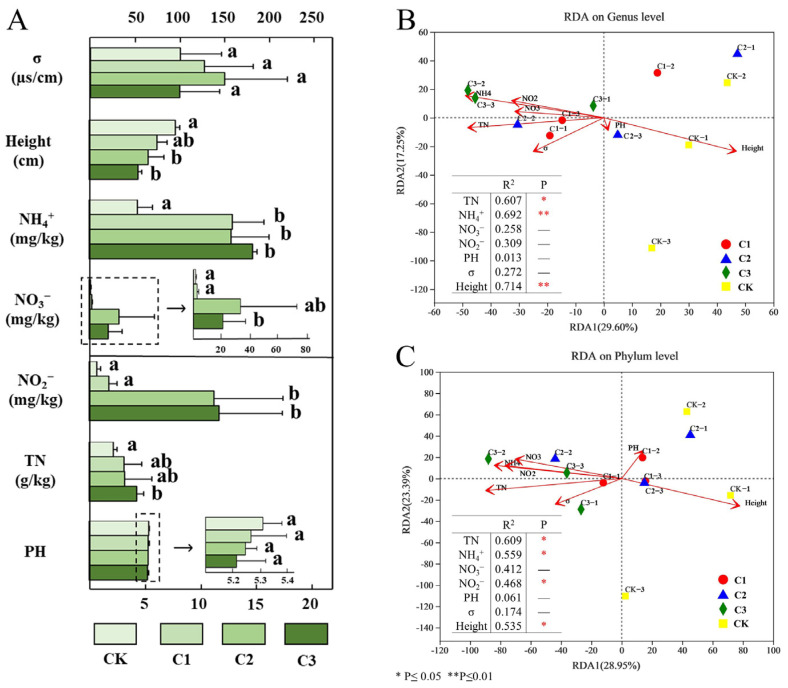
(**A**) Environmental factors under N addition. Error bars are based on the standard deviation of the means (*n* = 3). Letters are used to indicate a significant difference. (**B**) Redundancy analysis (RDA) of bacterial communities and environmental factors under genus level. (**C**) Redundancy analysis (RDA) of bacterial communities and environmental factors under phylum level. CK: CK−1, CK−2, CK−3, and 0 gN·m^−2^·a^−1^; C1: C1−1, C1−2, C1−3, and 6 gN·m^−2^·a^−1^; C2: C2−1, C2−2, C2−3, and 12 gN·m^−2^·a^−1^; C3: C3−1, C3−2, C3−3, and 24 gN·m^−2^·a^−1^. * Significant at the 0.05 level, ** Significant at the 0.01 level.

**Figure 5 microorganisms-11-01552-f005:**
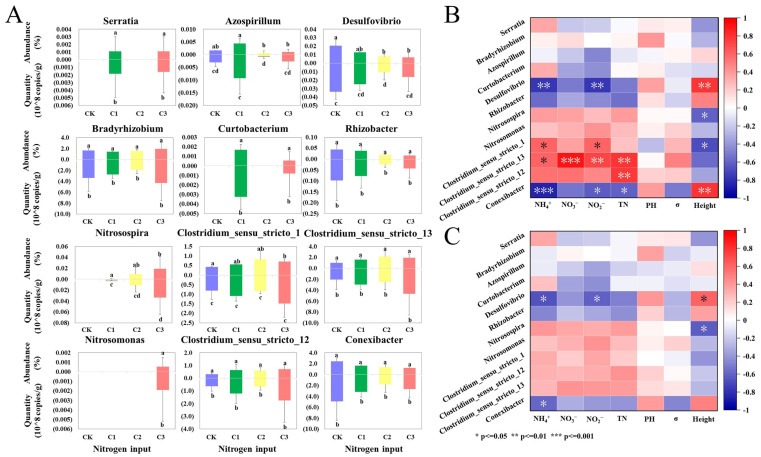
(**A**) Trend in relative abundance and absolute quantity of N fixing, nitrifying, and denitrifying bacteria under N addition. CK: 0 gN·m^−2^·a^−1^; C1: 6 gN·m^−2^·a^−1^; C2: 12 gN·m^−2^·a^−1^; C3: 24 gN·m^−2^·a^−1^. Error bars are based on the standard deviation of the means (*n* = 3). Letters are used to indicate a significant difference. (**B**) Correlation of environmental factors and relative abundance of N-cycling bacteria. (**C**) Correlation of environmental factors and absolute quantity of N-cycling bacteria. * Significant at the 0.05 level. ** Significant at the 0.01 level. *** Significant at the 0.001 level.

**Figure 6 microorganisms-11-01552-f006:**
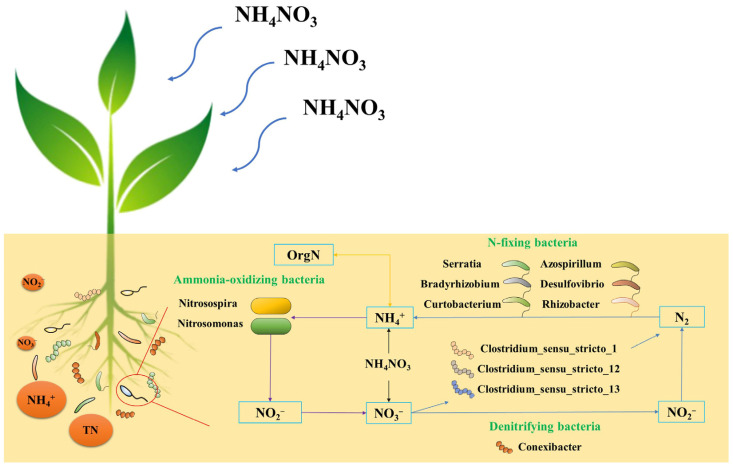
Schematic diagram of soil nitrogen cycle microorganisms. OrgN: organic nitrogen.

## Data Availability

Data are available on request.

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
