# Peer review of "Effects of Long-Term (17 Years) Nitrogen Input on Soil Bacterial Community in Sanjiang Plain: The Largest Marsh Wetland in China"

_microorganisms, 2023, doi:10.3390/microorganisms11061552_

Round 1
Reviewer 1 Report
The reviewed work is interesting. The Authors discuss the impact of long-term exogenous N input on microbial communities. They suggest that long-term exogenous N input can influence on soil function in the case of wetlands.
Lines 78-79: Latin names like Elaeagnus microphylla, Salix rosmarinifolia, Carex spp. should be written in italics
Line 89: The caption under the figure should be fuller. You should specify what is CK, C1, C2, C3 or provide a link to the methods
Line 172: Indexes Shannon and Simpson indicated only no significant differences of soil bacteria communities of different experimental variants
Line 179: Figure 2 The abbreviations C1, C3, etc. are used in the text, and the amounts of N are given in the figure. It would be preferable to choose one description in the text and in the figure.
Lines 185-186 ‘These dominant bacteria didn’t change significantly under different levels of long-term N input, indicating relatively stable populations and adaptability to environmental changes’ - I see at the end of this section, that is parts - 3.1. Diversity and composition of microbial communities
The same in the case of the line 189: ‘Long-term high-level N input was selective for wetland 189 soil bacterial species’
Line 317: The publication number should follow the name. Is ‘Wei…….[54]’ should be ‘Wei[54]….. The same in the case of the line 339: Cai et al.[65]
Author Response
Dear reviewer
Thank you for your recognition and comments on our manuscripts. We have studied comments carefully and have made correction which we hope meet with approval. Our responses to each comment were in red and the line number after the response refers to the manuscript of “microorganisms-2385570-all markers”. Please refer to the attachment for the main corrections in the paper and responses to comments. Thank you and best regards.

Reviewer 2 Report
The paper describes the increased nitrogen input from natural factors and human activities can harm marsh wetland health. The study examined the effects of long-term nitrogen input on the soil bacterial community, a key indicator of ecosystem health. High nitrogen levels reduced bacterial diversity and inhibited dominant microorganisms. Nitrogen influenced the soil microbial community, decreasing nitrogen-fixing microorganisms and increasing nitrifying and denitrifying microorganisms. This suggests that excess nitrogen inhibits wetland nitrogen fixation but enhances nitrification and denitrification processes and inform wetland conservation strategies.
The discussion adds depth to the paper and is excellent.
Several questions for discussion:
1. What is the effect on the fungal population? (Specified in the introduction)?
2. Which plants are interacting with the bacteria that will be affected? (line 240-241)
3. Did the depth of the soil affect the differences in the bacterial population?
4. What would happen if nitrogen were added several times - as can be done in swamp treatments?
5. What will the bacteria population be over time after adding the treatment, for example 1 day 5 days 20 days.
To make it easier for the reader, several suggestions:
1. Scheme of the nitrogen cycle and the bacteria associated with each stage.
2. Figure 1: Please check the richness. Alpha diversity- it’s not clear why there is line between the treatments. Expand the figure legend that will explain more and give more details.
3.Match the names of the treatments to the graphs - c1 -test? It is better to write the treatment itself and detail it in the legend of the illustration (Figures 2-4)
4. Instead of c11 write c1-1
No comments
Author Response
Dear reviewer
Thank you for your comments concerning our manuscript. Those questions and suggestions are all valuable and very helpful for revising and improving our paper. Our responses to each comment are shown in red, and the line number following the responses is the manuscript of "microorganisms-2385570-all markers". Please refer to the attachment for the main corrections in the paper and responses to comments. Thank you and best regards.

Reviewer 3 Report
The research is devoted to a significant study, the long-term application of nitrogen to soils, and the soil microbial community response. However, the authors have not described several issues in the article. First, soil types listed as “black soil, albic soil, meadow soil, and swamp soil” do not correspond to any soil classifications. Secondly, part of the article is based on a discussion of the composition of the soil microbial community, but from Fig 2 (community composition of soil bacteria at genus level data), it can be seen that about 30% of the community is unknown; therefore, it is not entirely correct to discuss the effect of nitrogen fertilizers on microbial diversity. Thirdly, it is not clear from the text of the article what g m–2 a–1 is, and for what reason was the 10–20 cm horizon studied, and not the entire root zone of the soil. Fourth, it is not clear from the text of the manuscript why after 17 years of nitrogen application, the difference between the N content of the control plot and the experimental plots was only 2 times, and the difference in total N content between experimental plots was not significant (Fig. 4a). In my opinion, in its present form, the manuscript cannot be published in the journal Microorganisms.
Author Response
Dear reviewer
Thank you for your four comments concerning the manuscript. We have carefully considered these questions and provided our answers. Our responses to each comment were in red and the line number after the response refers to the manuscript of “microorganisms-2385570-all markers”. Please refer to the attachment for the main corrections in the paper and responses to comments. Thank you and best regards.

Round 2
Reviewer 3 Report
Please arrange in italics the Latin names of plants and microorganisms throughout the text of the manuscript.
Author Response
Dear reviewer
Thank you for your comments and suggestions on our manuscripts. Our responses to your comments are shown below, and the line numbers following the responses are the manuscript of "microorganisms-2385570(2) ". Thank you and best regards.
Point 1: Please arrange in italics the Latin names of plants and microorganisms throughout the text of the manuscript.
Response 1: We have arranged the Latin names of plants and microorganisms in italics in the entire manuscript. (Lines 17-19, Lines 81-82, Line 118, Lines 200-201, Line 204, Lines 207-218, Lines 280-288, Lines 297-305, Lines 309-315, Lines 324-326, Line 329, Line 380, Line 382, Line 394, Lines 403-404, Line 418)